# Characterization of *Escherichia coli* in Dogs with Pyometra and the Influence of Diet on the Intestinal Colonization of Extraintestinal Pathogenic *E. coli* (ExPEC)

**DOI:** 10.3390/vetsci9050245

**Published:** 2022-05-22

**Authors:** Rafael Gariglio Clark Xavier, Paloma Helena Sanches da Silva, Hanna Dornelas Trindade, Gabriela Muniz Carvalho, Rafael Romero Nicolino, Patrícia Maria Coletto Freitas, Rodrigo Otávio Silveira Silva

**Affiliations:** Affiliation Veterinary School, Federal University of Minas Gerais, Antônio Carlos Avenue 6627, 31270-090 Belo Horizonte, Brazil; rafaelgariglio90@gmail.com (R.G.C.X.); palomasanches.vet@gmail.com (P.H.S.d.S.); nanadt@gmail.com (H.D.T.); munizcgabriela@gmail.com (G.M.C.); rafael.nicolino@gmail.com (R.R.N.); pcoletto@yahoo.com.br (P.M.C.F.)

**Keywords:** EnPEC, UPEC, RMDB, uterus, uterine, microbiota

## Abstract

Despite its high frequency and clinical relevance, the pathogenesis of canine pyometra remains poorly understood. The most accepted hypothesis is that bacteria involved ascend from the intestinal tract, causing the uterine infection. Extraintestinal pathogenic *Escherichia coli* (ExPEC) is the most frequent pathogen in canine pyometra, accounting for 57–100% of cases. The aim of the present study was to determine the frequency of phylogenetic groups and virulence factors in *E. coli* strains isolated from the uterine and rectal swabs of bitches with pyometra (*n* = 72) and from rectal swabs from healthy bitches fed commercial dry feed (*n* = 53) or a raw meat-based diet (RMBD; *n* = 38). A total of 512 strains of *E. coli* were isolated and divided into five categories according to the origin of the sample: 120 isolates from the uterine content of dogs with *E. coli* pyometra, 102 from the feces of bitches with *E. coli* pyometra, 75 from the feces of bitches without *E. coli* pyometra, 130 feces samples from healthy dogs fed commercial feed, and 85 feces samples from healthy dogs fed a raw meat-based diet. *E. coli* strains belonging to the B2 phylogroup and positive for virulence factor genes associated with adhesion (fimbriae type P [*papC*]) and production of toxins (α-hemolysin [*hlyA*] and uropathogenic specific protein [*usp*]) predominated in the uterine content and rectal swabs of bitches with *E. coli* pyometra. Interestingly, a lower growth rate of *E. coli* from the B2 phylogroup was observed in dogs fed a RMBD than in those fed commercial dry feed. The present study suggests that intestinal colonization by certain types of *E. coli* could be a risk factor for the occurrence of *E. coli* pyometra in bitches and that diet can influence intestinal colonization by such strains.

## 1. Introduction

Pyometra is the most frequently occurring reproductive disease in bitches, affecting up to 25% of uncastrated females [1,2]. The disease is characterized by bacterial infection of the uterus with local and systemic clinical manifestations that can lead to death [3,4,5]. However, despite its relevance, the pathogenesis of this disease remains poorly understood. It is believed that bacterial species causing pyometra ascend from the intestinal tract of females, causing infections [1,2,5,6].

Extraintestinal pathogenic *Escherichia coli* (ExPEC) is the most common pathogen involved in canine pyometra and has been reported in 57–100% of cases [1,2,6,7]. These isolates are phylogenetically and epidemiologically distinct from *E. coli* strains commonly found in intestinal commensals that cause diarrhea and other gastrointestinal disorders [8,9,10]. In canine pyometra, *E. coli* strains found in uterine contents are commonly associated with phylogroup B2 and less frequently with phylogroup D [11,12,13]. In contrast, commensal intestinal strains of *E. coli* in dogs are mostly classified into phylogenetic groups B1 and A [5,9,12]. In addition, *E. coli* recovered from pyometra have specific virulence factors, such as adhesins, toxins, iron acquisition systems, and protectins [7,14,15], which are commonly classified as endometrial pathogenic *E. coli* (EnPEC), a subgroup of the ExPEC pathotype. These virulence factors may confer a selective advantage over commensal strains [16], playing a key role in the development of canine pyometra [11,14,17] as well as in other extraintestinal infections in humans and animals [16,17,18].

Although the intestinal ascension of *E. coli* strains is currently the most accepted hypothesis in the pathogenesis of canine pyometra [1,2,7], no studies have evaluated the influence of the dog diet on the specific colonization of ExPEC strains. In the last decade, an increasing number of owners have been feeding their dogs and cats raw meat-based diets (RMBDs), instead of regular commercial dry feed [19]. Dogs fed an RMBD shed an increased amount of some pathogens in their feces, including *Salmonella* spp. and diarrheagenic *E. coli* [19,20,21]. However, specific virulence factors related to extraintestinal infections have not yet been investigated. The aim of this study was to determine the prevalence of phylogroups and virulence factors in *E. coli* isolates obtained from the uterine contents and feces of bitches with pyometra infection. In addition, we compared these *E. coli* isolates with those obtained from the feces of healthy dogs fed commercial dry feed or an RMBD to evaluate the possible influence of diet on colonization by *E. coli* strains.

## 2. Materials and Methods

### 2.1. Sampling

Three groups of bitches were sampled in the present study: dogs with pyometra (uterine and rectal swabs), healthy dogs fed commercial dry feed (rectal swab), and healthy dogs fed a RMBD (rectal swab). A total of 72 bitches with pyometra who underwent ovariohysterectomy (OH) surgery at the Veterinary Hospital of the Universidade Federal de Minas Gerais (VH-UFMG) between January 2017 and December 2020 were included. Immediately following surgery, aspiration puncture of the uterine contents was performed and a swab was introduced into the rectal ampulla of the bitches. The samples were refrigerated at 4 °C until processing for a maximum of 24 h. Rectal swabs from 91 healthy dogs were included, of which 53 were fed commercial dry feed, and 38 were fed a RMBD. The samples were kept in a cooler with ice packs and transported for processing within a maximum of 24 h. This study was approved by the Ethical Committee on Animal Use of UFMG (protocol no. 51/2015).

### 2.2. Isolation and Identification of E. coli

The uterine contents were plated on Mueller Hinton (MH) agar (Kasvi, Maharashtra, India) supplemented with equine blood (5%) and MacConkey (MC) agar (Difco, Franklin Lakes, NJ, USA), and the plates were incubated at 37 °C for 48 h under aerobiosis and anaerobiosis. Plating of rectal swab samples from female dogs subjected to OH and healthy dogs was performed on MC agar and incubated at 37 °C for 48 h under aerobiosis. For each clinical specimen, up to three lactose-fermenting colonies were subjected to species-specific polymerase chain reaction (PCR) to identify *E. coli* [22]. Strains not identified as *E. coli* was identified by matrix-assisted laser desorption/ionization-time of flight mass spectrometry (MALDI-ToF MS; Bruker Daltonics, Billerica, USA). A cutoff log score of 2 was used to validate the identification at the species level, as recommended by the manufacturer.

### 2.3. Characterization of E. coli

*E. coli* strains were subjected to PCR to determine phylogroups (A, B1, B2, C, D, E, F, or clade I) [23], and identify virulence genes corresponding to the ExPEC pathotype, namely, fimbriae type I (*fimH*), fimbriae type I central region (*focG*), fimbriae type P (*papC* and *papG*, allele II and III), fimbriae type S (*sfaS*), cytotoxic necrotizing factor type 1 (*cnf1*), uropathogenic specific protein (*usp*), α-hemolysin (*hlyA*), aerobactin (*iutA*), and serum resistance (*traT*) [18,24].

### 2.4. Statistical Analysis

The results were analyzed using EngineRoom software [25]. To analyze the association between *E. coli* phylogroups, virulence factors, and categorical variables related to the group origin of the samples (uterine content of bitches with *E. coli* pyometra, rectal swabs of bitches with *E. coli* pyometra, or without *E. coli* pyometra, and healthy dogs fed commercial dry feed or a RMBD) a multiple proportion comparison test was conducted. This test is based on the chi-square distribution and the pooled estimate of the population proportion to estimate the standard error of the test statistic. If a significant difference was found in the overall test, the pairwise comparisons method with Marascuillo procedure was used to identify the specific pairs of proportions which differ significantly. Statistical significance of the results was set at *p* ≤ 0.05 for the analyzed characteristics [26].

## 3. Results

### 3.1. E. coli Isolation

A total of 40 (56%) of the 72 dogs tested positive for *E. coli* in the uterine content; up to three colonies were obtained from each, totaling 120 *E. coli* strains, while 21 (29%) had only other pathogens, and no bacterial growth was seen in 11 (15%) (Table 1).

Up to three colonies of *E. coli* were obtained from rectal swabs of 59 bitches with pyometra, totaling 177 *E. coli* strains: 102 from dogs that tested positive for *E. coli* content (*E. coli* pyometra) and 75 from bitches that tested negative for *E. coli* (without *E. coli* pyometra) in the uterine contents.

From healthy bitches, at least one *E. coli* isolate was recovered from 91 dogs sampled, totaling 215 strains: 130 and 85 from dogs fed commercial feed or RMBD, respectively.

### 3.2. E. coli Phylogroups

Phylogroup B2 was the most common *E. coli* phylogroup detected in the uterine contents of bitches infected with *E. coli* pyometra (85%) and also in the rectal swab isolates of bitches with *E. coli* pyometra (58.8%), whereas B1 was most frequent in the rectal swabs of bitches without *E. coli* pyometra (41.3%). Bitches with *E. coli* pyometra showed a higher frequency of phylogroup B2 in the rectal swab than females without *E. coli* pyometra (*p* < 0.05). Phylogroup B2 was also the most frequent in *E. coli* isolates from rectal swabs of dogs fed commercial dry feed (34.6%), whereas B1 was the most common in dogs fed RMBD (34.1%). Dogs fed commercial dry feed showed a higher frequency of phylogroup B2 in rectal swabs than dogs fed RMBD (*p* < 0.05) (Table 2).

### 3.3. Frequency of Virulence Genes Associated with the ExPEC Pathotype

All the virulence genes tested were detected in *E. coli* isolates from all groups at different frequencies. Virulence genes associated with adhesion (papC) and toxin production (hlyA and usp) were more frequent in the rectal swabs of bitches with *E. coli* pyometra than in those without *E. coli* pyometra (*p* < 0.05). In addition, two virulence genes associated with adhesion (focG and sfaS) were more frequent in isolates from dogs fed commercial dry feed than in those from dogs fed RMBD (*p* < 0.05). In contrast, the serum resistance gene (traT) was found at a higher frequency in isolates from dogs fed RMBD than in those from dogs fed commercial dry feed (*p* < 0.05) (Table 3).

## 4. Discussion

As expected, *E. coli* was isolated from most of the uterine contents of dogs with pyometra. This result is in accordance with previous studies showing that *E. coli* is the main bacterium involved in pyometra [6,27,28].

Differentiation into phylogenetic groups and the detection of virulence factors have been widely used in studies on *E. coli*, helping to elucidate the epidemiology of infections and the colonization dynamics of these bacteria [5,23,29]. Previous studies have demonstrated that ExPEC strains isolated from canine pyometra tend to cluster mainly in phylogroup B2, whereas those isolated from the intestinal microbiota of healthy dogs cluster mainly in phylogenetic groups B1 and A [9,12,30]. In the present study, phylogroup B2 was the most frequent in the uterine contents of bitches, with clinical cases of pyometra caused by *E. coli* (Figure 1), corresponding to 85% of the isolates. This frequency is similar to that found in previous studies on pyometra, suggesting a high capacity of phylogroup B2 strains to colonize the canine uterus [7,11,31].

Although the pathogenesis of pyometra is poorly understood, previous studies have suggested that the intestine is the main source of *E. coli* strains that ascend into the uterus [1,2,7]. This study reinforces this hypothesis, as bitches with *E. coli* pyometra were more likely to harbor *E. coli* strains from phylogroup B2 in the rectal swab when compared to the group of bitches without *E. coli* pyometra. This finding indicates that intestinal colonization by *E. coli* from phylogroup B2 increases the risk of pyometra in female dogs.

Another interesting aspect of ExPEC is the presence of certain virulence factors that enable infection at different locations [5]. Virulence factors that promote adhesion and colonization, especially fimbriae, are considered to be of great relevance for the establishment of *E. coli* infections in the canine uterus [18,32,33]. Previous studies demonstrated that simple inactivation of some adhesins, such as type 1 (*fim*), P (*papGIII*), and S (*sfa/foc*) fimbriae, results in a considerable reduction in bacterial binding to cell lines of the canine endometrium, reinforcing the importance of these factors in the pathogenesis of the disease [34]. In the present study, four adhesin-encoding virulence genes were found more frequently in *E. coli* samples obtained from the uterine contents, similar to the findings of previous studies [11,18,31]. This finding reinforces the hypothesis that some adherent virulence factors are associated with pyometra caused by *E. coli* in female dogs. It is noteworthy that the gene encoding type P fimbriae (*papC*), which is considered important for the adhesion and colonization of *E. coli* in the canine endometrium [6,7], was found in 55% of the isolates from the uterine contents. This frequency is similar to that identified in other studies on canine pyometra isolates [5,18]. In addition, strains isolated from the rectal swabs of bitches with *E. coli* pyometra were more commonly positive for the type P fimbriae gene (*papC*) than strains isolated from the rectal swabs of dogs without *E. coli* pyometra. Notably, the frequency of *papC*-positive *E. coli* strains in dogs without *E. coli* pyometra was similar to that reported in a previous study on *E. coli* from rectal swabs of healthy dogs [17].

Although *E. coli* is known to be the main bacterium involved in pyometra [1,2], and recent studies have suggested that diet can influence *E. coli* colonization [8,35,36], current studies have evaluated how different diets would affect the frequency of ExPEC in bitches. In the present study, the *papC* gene showed no statistical difference between the groups of healthy bitches under different types of feeding. In contrast, the genes encoding type 1 adhesin (*focG*) and S (*sfaS*) fimbriae were found less frequently in *E. coli* strains recovered from dogs fed RMBD. These adhesins are considered important in the pathogenesis of canine pyometra [11,32,37]. However, it is important to note that the frequency of these two adhesin-encoding genes was similar in strains isolated from rectal swabs of dogs with or without *E. coli* pyometra, raising doubts regarding the role of these virulence factors in disease development.

Previous studies have indicated that ExPEC obtained from the uterine content of bitches with pyometra commonly expresses genes encoding toxins that may provide a selective advantage [1,2,7,38]. We observed that all toxin-coding virulence genes were found more frequently in *E. coli* samples obtained from the uterine content, which is in agreement with previous studies [6,18,31], which reinforces the hypothesis that, in addition to adhesins, ExPEC toxin virulence factors are associated with the occurrence of *E. coli* pyometra. Among the *E. coli* isolates from rectal swabs, the α-hemolysin (*hlyA*) toxin, which is capable of lysing erythrocytes and leukocytes [31,38,39], was found more frequently in strains isolated from bitches with *E. coli* pyometra than in strains isolated from rectal swabs of dogs without *E. coli* pyometra. Additionally, the uropathogenic specific protein (*usp*), which acts as a bacteriocin and assists in the migration of strains into the bloodstream [18,33,40], was more frequent in strains isolated from the rectal swabs of bitches with *E. coli* pyometra than in strains isolated from dogs without *E. coli* pyometra.

ExPEC obtained from the uterine contents of bitches with pyometra is commonly positive for the aerobactin gene (*iutA*), a virulence factor responsible for iron acquisition [5,38], and for the serum resistance gene (*traT*), a virulence factor associated with the inhibition of the immune response of the host in cases of translocation of the pathogen into the bloodstream [31,39]. In the present study, both virulence genes were detected in all groups, and the frequency was similar among *E. coli* strains obtained from uterine content and rectal swabs from bitches with *E. coli* and without *E. coli* pyometra. In contrast, *traT* was more frequently detected in *E. coli* strains from rectal swabs of dogs fed RMBD than in those fed commercial dry feed.

Research on phylogroups and virulence factors of *E. coli* from different origins has increased over the last few years, but many gaps remain, mostly regarding *E. coli* colonization and infection in dogs [1,7]. In the present study, we demonstrated that, compared to dogs without *E. coli* pyometra, dogs with *E. coli* pyometra are more likely to be colonized by *E. coli* from phylogroup B2, which is positive for specific virulence genes, including type 1 adhesin (*papC*) and two toxins (*hlyA* and *usp*). These results suggest that colonization by these strains is a risk factor for canine pyometra caused by *E. coli*. Based on these results, we sampled two groups of healthy dogs under different diets to evaluate whether dietary habits altered the intestinal microbiota and further established *E. coli* in the B2 phylogroup. Our results suggest that dogs fed RMBD are less frequently colonized by *E. coli* strains from phylogroup B2, raising the hypothesis that diet can be a risk factor for the occurrence of *E. coli* pyometra, which is the main bacterium responsible for this disease [6,7,41].

Importantly, several studies have indicated public health risks associated with RMBD, such as greater fecal shedding of pathogenic and zoonotic microorganisms, which is a potential risk to animal and human health [19,42]. Therefore, several health agencies have released statements that discourage the inclusion of raw or undercooked animal protein in dog diets [42]. We believe that the results of this study will motivate future evaluations of different diets for dogs that aim to reduce the colonization of ExPEC, however, this study should not be considered as a motivation for the adoption of RMDB, owing to the known risks of this practice.

## 5. Conclusions

The present study demonstrated the high frequency of *E. coli* strains belonging to phylogroup B2 and carrying virulence factors associated with ExPEC in isolates from the uterine contents of bitches with pyometra. In addition, this study found a higher frequency of these strains in the intestinal microbiota of bitches with *E. coli* pyometra than in bitches without *E. coli* pyometra, suggesting that intestinal colonization by these strains could be a risk factor for the occurrence of *E. coli* pyometra in dogs. Interestingly, when evaluating the intestinal microbiota of dogs on different types of diets, the present study found a lower frequency of such strains in the intestinal microbiota of dogs subjected to a RMBD than in dogs who consumed commercial dry feed, suggesting that future studies on diet modulation affecting intestinal colonization could find mechanisms to prevent and control *E. coli* pyometra in dogs.

## Figures and Tables

**Figure 1 vetsci-09-00245-f001:**
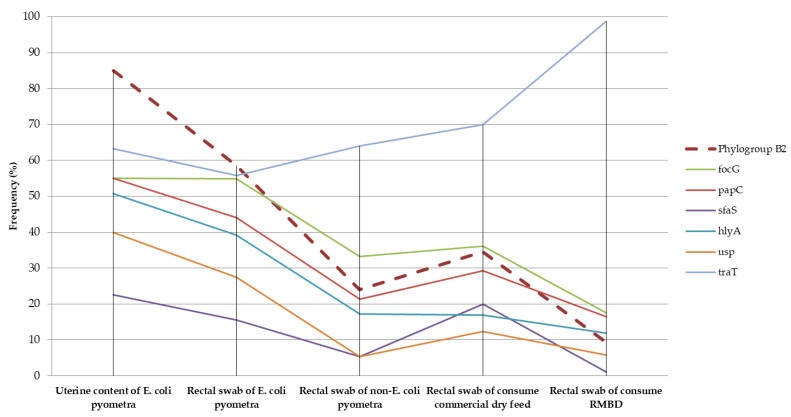
Frequency of the phylogroup B2 and the main virulence factors identified in *E. coli* isolated from the uterine content, rectal swabs of bitches with pyometra and rectal swabs of healthy dogs fed commercial dry feed and raw meat-based diet (RMBD).

**Table 1 vetsci-09-00245-t001:** Bacterial species isolated from the uterus in bitches with pyometra.

Organism	Total Cases (%)
*Escherichia coli*	40 (56)
*Staphylococcus* sp.	6 (8)
*Streptococcus* sp.	6 (8)
*Enterobacter* sp.	2 (3)
*Enterococcus* sp.	2 (3)
*Klebsiella pneumoniae*	2 (3)
*Proteus mirabilis*	2 (3)
*Pseudomonas aeruginosa*	1 (1)
No growth	11 (15)
Total	72 (100)

**Table 2 vetsci-09-00245-t002:** Number of isolates and frequency of *E. coli* phylogroups identified in the uterine content, rectal swabs of bitches with pyometra and rectal swabs of healthy dogs.

	Bitches with Pyometra			Healthy Dogs	
	Uterine Content	Rectal Swab			
Phylogroup	*E. coli* Pyometra	*E. coli* Pyometra	Non-*E. coli* Pyometra	Consume Commercial Dry Feed	Consume RMBD
A	0	2 (1.9%)	5 (6.6%)	4 (3%)	6 (7%)
B1	3 (2.5%)	22 (21.5%)	31 (41.3%)	35 (26.9%)	29 (34.1%)
B2	102 (85%)	60 (58.8%) ^a^	18 (24%)	45 (34.6%) ^b^	8 (9.4%)
C	0	4 (3.9%)	0	16 (12.3%)	11 (12.9%)
D	0	0	2 (2.6%)	1 (0.7%)	0
E	6 (5%)	7 (6.8%)	6 (8%)	10 (7.6%)	20 (23.5%)
F	3 (2.5%)	3 (2.9%)	10 (13.3%)	11 (8.4%)	10 (11.7%)
*E*. clades—clade I	0	0	0	3 (2.3%)	0
Not classified	6 (5%)	4 (3.9%)	3 (4%)	5 (3.8%)	1 (1.1%)
Total	120	102	75	130	85

^a^ Samples with statistical difference when comparing strains obtained from the rectal swab of bitches with *E. coli* pyometra and bitches with non-*E. coli* pyometra. ^b^ Samples with statistical difference when comparing strains obtained from the rectal swabs of dogs fed commercial dry food and dogs fed RMBD.

**Table 3 vetsci-09-00245-t003:** Number of isolates and frequency of *E. coli* virulence genes identified in the uterine content, rectal swabs of bitches with pyometra and rectal swabs of healthy dogs.

	Bitches with Pyometra			Healthy Dogs	
	Uterine Content	Rectal Swab			
Virulence Genes	*E. coli* Pyometra	*E. coli* Pyometra	Non-*E. coli* Pyometra	Consume Commercial Dry Feed	Consume RMBD
**Adhesion**					
*fimH*	120 (100%)	102 (100%)	75 (100%)	128 (98.4%)	83 (97.6%)
*focG*	66 (55%)	56 (54.9%)	25 (33.3%)	47 (36.2%) ^b^	15 (17.6%)
*papC*	66 (55%)	45 (44.1%) ^a^	16 (21.3%)	38 (29.2%)	14 (16.4%)
*papG*	58 (48.3%)	36 (35.2%)	13 (17.3%)	74 (56.9%)	37 (43.5%)
*sfaS*	27 (22.5%)	16 (15.6%)	4 (5.3%)	26 (20%) ^b^	1 (1.1%)
**Toxins**					
*cnf1*	50 (41.6%)	33 (32.3%)	14 (18.6%)	20 (15.3%)	9 (10.5%)
*hlyA*	61 (50.8%)	40 (39.2%) ^a^	13 (17.3%)	22 (16.9%)	10 (11.7%)
*usp*	48 (40%)	28 (27.4%) ^a^	4 (5.3%)	16 (12.3%)	5 (5.8%)
**Iron acquisition**					
*iutA*	43 (35.8%)	39 (38.2%)	26 (34.6%)	103 (79.2%)	74 (87%)
**Serum resistance**					
*traT*	76 (63.3%)	57 (55.8%)	48 (64%)	91 (70%)	84 (98.8%) ^b^
Total	120	102	75	130	85

^a^ Samples with statistical difference when comparing strains obtained from the rectal swab of bitches with *E. coli* pyometra and bitches with non-*E. coli* pyometra. ^b^ Samples with statistical difference when comparing strains obtained from the rectal swabs of dogs that consume commercial dry food and dogs that consume RMBD.

## Data Availability

Data is available upon reasonable request.

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
