# Peer review of "Characterization of Escherichia coli in Dogs with Pyometra and the Influence of Diet on the Intestinal Colonization of Extraintestinal Pathogenic E. coli (ExPEC)"

_vetsci, 2022, doi:10.3390/vetsci9050245_

Round 1

Reviewer 1 Report

The manuscript entitled "Characterization of Escherichia coli in dogs with pyometra and the influence of diet on the intestinal colonization of extraintestinal pathogenic E. coli (ExPEC)" deals with the isolation and characterization of E.coli strains isolated from uterine and rectal samples collected from both bitches with pyometra and healthy bitches. The article is well written and gives interesting results on  the frequency of the  main phylogenetic extra intestinal E. coli groups and their  virulence factors in both diseased and healthy dogs. Furthermore, in healthy dogs the influence of diffent diets on the colonization of  intestinal E.coli was evaluated. However, improvements are necessary for a better scientific rigor, especially in the material and methods section. 

Matherials and methods

Particularly, materials and methods should be described in more detail in the paragraph "2.2. Isolation, identification, and characterization of E. coli".  In effect, I would suggest dividing this paragraph in two: in one the authors should better  describe the techniques and the methods used for the  isolation and identification of the strains, in the other they should focus on the strains characterization. 

Precisely:  

lines 81-82 authors report that they used both  MH addiotioned with equine blood  and MC for the isolation of the strains. Why did the use MH? Generally, this medium is used for the antimicrobial susceptibility testing.  Did the authors evaluate the antimicrobial resistance profiles of the isolate E.coli strains? Otherwise, if the authors wanted to the dected the presence of other bacteria together with E.coli, such as Gram- positive bacteria they should have used Columbia CNA agar (a selective medium for Gram- positive). In addition,  if the authors wanted to evaluate phenotypically the ability of E.coli strains to induce hemolysis, they could have used Tryptic Soy agar. 

Please specify the reason why you used MH. 

Lines 86-87: did the authors identify the isolated strains only by molecular methods? Did you not perform screening test (Gram staining, catalase or oxidase test) or other identification systems before the PCR ? 

Results:

Lines 107:  authors reported that 21 (29%) sample had only other pathogens.  How they can state this? How did they identify them? Which bacteria were? Be precise about it. 

A minor English revision of the manuscript is required. 

Author Response

The manuscript entitled "Characterization of Escherichia coli in dogs with pyometra and the influence of diet on the intestinal colonization of extraintestinal pathogenic E. coli (ExPEC)" deals with the isolation and characterization of E.coli strains isolated from uterine and rectal samples collected from both bitches with pyometra and healthy bitches. The article is well written and gives interesting results on the frequency of the  main phylogenetic extra intestinal E. coli groups and their virulence factors in both diseased and healthy dogs. Furthermore, in healthy dogs the influence of diffent diets on the colonization of intestinal E.coli was evaluated.

Authors: Thank you for this comment.

However, improvements are necessary for a better scientific rigor, especially in the material and methods section. Particularly, materials and methods should be described in more detail in the paragraph "2.2. Isolation, identification, and characterization of E. coli".  In effect, I would suggest dividing this paragraph in two: in one the authors should better describe the techniques and the methods used for the isolation and identification of the strains, in the other they should focus on the strains characterization.

Authors: The whole material and methods were revised and paragraph 2.2 is now described in more details.

lines 81-82 authors report that they used both MH addiotioned with equine blood  and MC for the isolation of the strains. Why did the use MH? Generally, this medium is used for the antimicrobial susceptibility testing.  Did the authors evaluate the antimicrobial resistance profiles of the isolate E.coli strains? Otherwise, if the authors wanted to the dected the presence of other bacteria together with E.coli, such as Gram- positive bacteria they should have used Columbia CNA agar (a selective medium for Gram- positive). In addition,  if the authors wanted to evaluate phenotypically the ability of E.coli strains to induce hemolysis, they could have used Tryptic Soy agar. Please specify the reason why you used MH.

Authors: We used MH medium supplemented with blood in aerobic and anaerobic conditions as it is very rich medium, useful for isolation of most gram-positive and gram-negative bacteria to associated with pyometra. Susceptibility to antimicrobials and the ability of E. coli isolates to induce hemolysis, although relevant and interesting, were not performed in the present study.

Lines 86-87: did the authors identify the isolated strains only by molecular methods? Did you not perform screening test (Gram staining, catalase or oxidase test) or other identification systems before the PCR ?

Authors: The colonies were submitted to E. coli species-specific PCR, the negative were then submitted to the MALDI-ToF technique to identify the bacterial species. We do not use biochemical techniques.

Lines 107:  authors reported that 21 (29%) sample had only other pathogens.  How they can state this? How did they identify them? Which bacteria were? Be precise about it.

Authors: Bacterial identification was performed using species-specific PCR for E. coli and MALDI-ToF for other bacterial species. A new table was added in article with the suggested details.

A minor English revision of the manuscript is required.

Authors: The paper was again submitted to an English proofreading

Reviewer 2 Report

The study investigated the frequency of E coli phylogroups in canine pyometra and analyzed the virulence factors associated with ExPEC in those cases. The study suggest that intestinal E coli infection could be a risk factor for canine pyometra, which is an important finding. It also investigated the effect of diet on gut microbiota in dogs. Overall, this study has provided critical information regarding ExPEC in bitches with pyometra and the influence of diet on ExPEC frequency and virulence factors.

Author Response

Reviewer 2

The study investigated the frequency of E coli phylogroups in canine pyometra and analyzed the virulence factors associated with ExPEC in those cases. The study suggest that intestinal E coli infection could be a risk factor for canine pyometra, which is an important finding. It also investigated the effect of diet on gut microbiota in dogs. Overall, this study has provided critical information regarding ExPEC in bitches with pyometra and the influence of diet on ExPEC frequency and virulence factors.

Authors: Thank you for this comment.

Reviewer 3 Report

This manuscript is well-written and interesting, however, this manuscript has some concerns.

  1. line 13-15, line 37-39, these sentences are exact duplicate
  2. Figure 1, provide a high-resolution figure
  3. line 182, in Discussion, avoid citing tables and figures, delete "(Table) 1"
  4. line 202, delete "p<0.05 Table2"
  5. line 211, delete "Table 2", also the same in line 225, 229, 238....
  6. line 241, add reference(s)

Author Response

Reviewer 3

This manuscript is well-written and interesting, however, this manuscript has some concerns.

Authors: Thank you for this comment.

line 13-15, line 37-39, these sentences are exact duplicate

Authors: Revised as suggested.

Figure 1, provide a high-resolution figure

Authors: Revised as suggested.

line 182, in Discussion, avoid citing tables and figures, delete "(Table) 1"

Authors: Revised as suggested.

line 202, delete "p<0.05 Table2"

Authors: Revised as suggested.

line 211, delete "Table 2", also the same in line 225, 229, 238....

Authors: Revised as suggested.

line 241, add reference(s)

Authors: Revised as suggested.

Round 2

Reviewer 1 Report

Dear authors,

thank you for your answers. The revisions made to the manuscript  improved it. Besides this, i noticed a minor revison to do (text editing):  Lines 112- 113 reports E.coli in italics. 

However, i wanted to clarify that I do not  completely agree in using MH supplemented with equine blood for the isolation of bacterial strains from uterine contents.  This medium is only  recommended  by EUCAST for antimicrobial susceptibility testing for fastidious organisms.   Authors could have  used other blood-enriched media recommended for the isolation of fastidious bacteria, such as brain heart infusion agar with 10% sheep blood...  

Author Response

Dear authors, thank you for your answers. The revisions made to the manuscript  improved it. Besides this, i noticed a minor revison to do (text editing):  Lines 112- 113 reports E.coli in italics. 

Authors: Revised. Thanks for being so careful and fast with the revision.

However, I wanted to clarify that I do not  completely agree in using MH supplemented with equine blood for the isolation of bacterial strains from uterine contents.  This medium is only  recommended  by EUCAST for antimicrobial susceptibility testing for fastidious organisms.   Authors could have  used other blood-enriched media recommended for the isolation of fastidious bacteria, such as brain heart infusion agar with 10% sheep blood.

Authors: After reviewing the literature, we acknowledge that we could have used other media instead of MH. This suggestion will be considered for future studies, thanks. Anyway, we believe that MH supplemented with blood, although not as rich as BHI, was enought to meet our objectives in the present study once our focusing was mostly on E. coli.